# Tumor Radiosensitization by Gene Electrotransfer-Mediated Double Targeting of Tumor Vasculature

**DOI:** 10.3390/ijms24032755

**Published:** 2023-02-01

**Authors:** Monika Savarin, Katarina Znidar, Gregor Sersa, Tilen Komel, Maja Cemazar, Urska Kamensek

**Affiliations:** 1Department of Experimental Oncology, Institute of Oncology Ljubljana, 1000 Ljubljana, Slovenia; 2Faculty of Health Sciences, University of Primorska, 6310 Izola, Slovenia; 3Faculty of Health Sciences, University of Ljubljana, 1000 Ljubljana, Slovenia; 4Faculty of Medicine, University of Ljubljana, 1000 Ljubljana, Slovenia; 5Biotechnical Faculty, University of Ljubljana, 1000 Ljubljana, Slovenia

**Keywords:** gene electrotransfer, silencing plasmid, antivascular shRNA, CD105, CD1046, radiosensitization, DNA sensors, murine mammary adenocarcinoma TS/A

## Abstract

Targeting the tumor vasculature through specific endothelial cell markers involved in different signaling pathways represents a promising tool for tumor radiosensitization. Two prominent targets are endoglin (CD105), a transforming growth factor β co-receptor, and the melanoma cell adhesion molecule (CD1046), present also on many tumors. In our recent in vitro study, we constructed and evaluated a plasmid for simultaneous silencing of these two targets. In the current study, our aim was to explore the therapeutic potential of gene electrotransfer-mediated delivery of this new plasmid in vivo, and to elucidate the effects of combined therapy with tumor irradiation. The antitumor effect was evaluated by determination of tumor growth delay and proportion of tumor free mice in the syngeneic murine mammary adenocarcinoma tumor model TS/A. Histological analysis of tumors (vascularization, proliferation, hypoxia, necrosis, apoptosis and infiltration of immune cells) was performed to evaluate the therapeutic mechanisms. Additionally, potential activation of the immune response was evaluated by determining the induction of DNA sensor STING and selected pro-inflammatory cytokines using qRT-PCR. The results point to a significant radiosensitization and a good therapeutic potential of this gene therapy approach in an otherwise radioresistant and immunologically cold TS/A tumor model, making it a promising novel treatment modality for a wide range of tumors.

## 1. Introduction

To grow beyond a limited size, solid tumors require a proper vasculature that grants oxygen, nutrients, and waste disposal. Therefore, activation of angiogenesis is needed to maintain the proliferation of tumor cells and their dissemination to distant sites. In recent decades, this therapeutic area has been extensively researched. Hypoxia, followed by a lack of nutrients, triggers the angiogenic switch and promotes the expression of inflammatory signals and cytokines, as well as the transcription of genes mandatory for angiogenesis, such as vascular endothelial growth factor (VEGF) and platelet-derived growth factor (PDGF) [1]. Antiangiogenic therapies using antibodies or tyrosine kinase inhibitors have been approved to treat several types of cancer. Despite the ever-growing list of FDA-approved drugs, the efficacy of the therapies results in short-term effectiveness, leading to resistance and modest survival benefits [2]. Therefore, targeting the tumor vasculature through specific endothelial cell markers involved in different signaling pathways and with different therapeutic approaches represents a promising tool for cancer treatment.

One of the promising targets is CD105 (endoglin), a transforming growth factor coreceptor that is crucial in developmental biology and tumor angiogenesis. It is highly expressed on tumor vessels and correlates with poor survival prognosis. The endoglin neutralizing antibody (TRC105; carotoximab), after being tested in many preclinical cancer models, has entered clinical studies (phase I–III) for cancer patients as mono- or combination therapy [3]. Despite very promising preclinical studies, pivotal trials for TRC105 did not exert clinical benefits. Another notable target is CD146 (melanoma cell adhesion molecule). It is an adhesion molecule and a VEGF-R2 coreceptor that promotes the transcription of many proangiogenic factors. CD146 is also expressed in many tumors, and its levels correlate with aggressiveness and invasiveness; therefore, it is also used as a marker for poor prognosis. An antibody targeting only CD146 expressed in tumor cells has already been tested in preclinical studies, demonstrating its effectiveness in reducing proliferation, migration, and tube formation in vitro and tumor growth and metastasis formation in vivo [4,5,6,7,8].

The main disadvantages of the therapeutic approaches described above are the instability of the antibodies and their short time of action. Therefore, it is reasonable to aim at specific markers with more sustainable, local, and specific approaches, which can be obtained using gene electrotransfer (GET). This is a safe nonviral gene therapy approach, which has demonstrated its efficacy in many preclinical studies and has also reached clinical application [9,10]. In oncology, several clinical trials are using plasmid DNA encoding different therapeutic molecules, such as interleukin 12 [11,12,13,14,15] or tumor-associated antigens [11,13,15,16,17,18].

Plasmids can also be used to express shRNA for gene silencing. Studies have shown that plasmid DNA allows longer and more stable silencing than siRNA [19,20]. It is also easier to produce and is more stable and resistant to nucleolytic degradation than RNA alone. Furthermore, an important advantage of plasmids is their capability to accommodate large genetic payloads, allowing co-transfection of multiple plasmids or transfection of larger polycistronic plasmids [21,22]. One possible limitation of plasmid DNA is the antibiotic resistance gene present in the backbone of conventional plasmids that is needed for their production in bacteria [23]. However, this can be circumvented with several new technologies, one of which is operator repressor titration (ORT) technology, which was also successfully implemented by our research group [24,25,26].

In our previous studies, we have shown that GET of plasmids encoding shRNA against either CD105 or CD146 results in significant vascular reduction and pronounced antitumor effects in several tumor models [27,28,29,30,31,32]. By combining this therapy with irradiation, the therapeutic effect was further improved [29,30,33], most likely due to the normalization of tumor vasculature that promotes better oxygenation status of the tumor and, therefore, tumor radiosensitization [34,35]. Additionally, the combined therapy can also promote activation of the immune system since many mice remained complete responders after secondary tumor challenge [30,33]. However, some complete responders were also obtained in the groups using plasmids devoid of therapeutic genes, indicating the action of the foreign plasmid DNA introduced via GET and DNA released from the cells after therapy-induced damage, which can act as pathogen-associated molecular patterns (PAMPs) or danger-associated molecular patterns (DAMPs), respectively [36,37,38,39]. These can activate cytosolic pattern recognition receptors (cytosolic DNA sensors), one of importance being the stimulator of interferon genes (STING), which triggers type I interferons and pro-inflammatory cytokine production. In our previous studies, we demonstrated activation of DNA sensors after GET and after irradiation (IR) of tumors separately; however, we did not determine this activation after our combined treatment with GET of shRNA-encoding plasmids and IR [36,37].

In our recent in vitro study, we evaluated a newly constructed plasmid for simultaneous GET-mediated silencing of CD105 and CD146, which was devoid of an antibiotic resistance gene [25]. The study confirmed the functionality of the plasmid, as both of the targets were successfully silenced. This prompted a new study to explore the therapeutic potential in vivo and to elucidate the effects of combined therapy with tumor IR. The results of this study point to significant radiosensitization and immunostimulatory effectiveness of this gene therapy approach in an otherwise radioresistant and immunologically cold murine mammary adenocarcinoma TS/A tumor model [40,41]. Altogether, a good therapeutic potential of the simultaneous silencing of the two targets responsible for distinct angiogenic pathways in combination with irradiation was demonstrated, making it a promising novel treatment modality for a wide range of tumors.

## 2. Results

### 2.1. Dual Silencing of CD105 and CD146 Promotes Radiosensitization of TS/A Tumors

Injection of plasmids, application of electric pules (EP) or IR alone, did not affect tumor growth delay (TGD). GET of the dual silencing plasmid pDual resulted in 11.3 days of TGD, which was statistically significantly longer than after GET of the control plasmid pEmpty (3.5 days). When GET of either of the plasmids or even EP alone was combined with tumor IR, the TGDs were statistically significantly prolonged. The longest TGD, i.e., 23.9 days, was obtained after combining GET of pDual with IR, which was significantly longer than that after GET pDual monotherapy (13 days) (Table 1, Figure 1a,b).

Additionally, 50% of mice were cured in this therapeutic group, although this was not statistically significant compared with the EP + IR, GET pDual, and GET pEmpty + IR groups, where there were also some complete responders (1–2 mice) (Table 1, Figure 1c). The tumor-free mice were challenged with the injection of tumor cells 100 days from the beginning of the treatment to determine possible induction of immune memory. In general, mice were not resistant to the secondary challenge, with the exception of one mouse in the GET pDual + IR group.

No body weight loss over 10% or any other side effects were observed, except for temporary hair loss in the irradiated area without skin desquamation, proving the safety of the treatment.

### 2.2. Histological Analyses Indicate Vascular Targeted Effects Combined with Immune Activation

Immunohistochemistry (IHC) analyses of the tumor samples obtained on day 6 after the beginning of the therapy were performed to evaluate the levels of tumor vascularization, proliferation, hypoxia, apoptosis, immune infiltration (granzyme B-positive cells), and necrosis (Figure 2a). In general, regression of the tumor vasculature was observed in the therapeutic groups (Figure 2b). The highest and statistically significant results were observed in the GET pDual + IR group, where the vascularization was reduced to 17%. The same trend was also observed in proliferation, which was the most reduced after GET of either of the plasmids (pEmpty, pDual) in combination with IR, i.e., to 33.5% or 29%, respectively (Figure 2c). On the other hand, the percentage of hypoxia, apoptosis, necrosis, and granzyme B-positive cells was elevated after GET monotherapies and in combination with IR. Specifically, hypoxia levels were significantly increased compared with the control and IR alone in all groups where EP was used either alone or in combination with plasmids (GET) or in combination with IR (Figure 2d). The highest elevation (62.4% of hypoxic cells) was achieved after the GET of the therapeutic plasmid combined with IR. Very similar results were also obtained for necrosis (Figure 2e). Correlating to tumor size, the percentage of the necrotic area was higher in the groups receiving GET of either of plasmids alone or combined with IR. The induction of apoptosis was moderate but still statistically significant compared with the control and IR alone in groups combining GET of either of the plasmids with IR; however, the highest induction was achieved after GET of the therapeutic plasmid combined with IR (19.1% of apoptotic cells, Figure 2f). Regarding the infiltration of immune cells, the number of granzyme B-positive cells was significantly increased in groups receiving GET of either plasmid (Figure 2g). When GET was further combined with IR, the number of positive cells was even more pronounced (23.3% of granzyme B-positive cells).

### 2.3. DNA Sensor Sting and Proinflammatory Cytokine Levels Indicate Induction of Immune Response

The expression levels of the cytokines Il1β, Ifn-β1 and Tnf-α and the DNA sensor Sting were measured on day 6 after GET of the pDual in TS/A tumors (Figure 2h). All the cytokines were detected in the control, untreated groups of a TS/A tumor model. The average CT value (n = 6) was 27.7 for Il1β, 33.8 for Ifn- β1 and 28.5 for Tnf-α. All cytokines were significantly increased in the group where GET of the control plasmid (pEmpty) or therapeutic plasmid (pDual) was combined with IR (GET pEmpty + IR and GET pDual + IR). A 20.3-fold and 10.4-fold increase in the expression of the proinflammatory cytokine Il1β, a 75-fold and 83.4-fold increase in the expression of the cytokine Ifn-β1 and a 10.4-fold and 10.5-fold increase in the expression of the proinflammatory cytokine Tnf-α were observed after GET pEmpty + IR and GET pDual + IR, respectively. Ifn-β1 was also significantly increased (68.2-fold) after GET of pEmpty. The DNA sensor Sting was detected at a CT level of 25.5 in the control tumors. After GET of pEmpty and pDual in combination with IR, a statistically significant 2.6-fold increase in STING expression was detected.

## 3. Discussion

The study aimed to evaluate the therapeutic effectiveness of gene therapy with a new plasmid DNA devoid of an antibiotic resistance gene, simultaneously silencing two independent targets, CD105 and CD146, which both have demonstrated great potential in preclinical and clinical studies. Gene therapy was performed using GET and was further combined with single-dose IR (15 Gy) in a murine mammary adenocarcinoma TS/A tumor model that does not express either of the targeted markers to distinguish between direct effects on the vasculature and tumor cells [33,42]. The results showed radiosensitization of the treated tumors, which resulted in prolonged tumor growth delay and up to 50% of tumor-free mice. Histological analyses demonstrated a significant decrease in tumor vascularization, mainly due to GET, which in combination with IR reduced the proliferation of tumor cells, resulting in an increase in necrosis, hypoxia, apoptosis and infiltration of immune cells. The treatment was accompanied by the activation of DNA sensors and proinflammatory cytokines, which were upregulated in groups combining GET and IR, indicating activation of the immune system. However, since the tumors regrew after the secondary challenge, with the exception of one mouse in the therapeutic group, the induced innate immune response was obviously not converted to specific immunological memory. Generally, these findings are in line with the results of our previous study [29] that was also performed in an immunologically cold TS/A tumor model using plasmids targeting only one vascular marker, either CD105 [29,43] or CD146 [33]. However, with the current therapy, targeting both vascular markers simultaneously, overall tumor growth delays and complete responders were more pronounced.

Combining vascular-targeted therapies and irradiation was proven to be an effective treatment modality since by regulating tumor oxygenation, the effectiveness of irradiation can be enhanced. Namely, antiangiogenic agents can improve tumor oxygenation status through normalization of tumor vasculature, vessel depletion and even immune activation [2,34], whereas vascular-disrupting agents can promote a more oxygenated tumor rim through disruption of already formed tumor vasculature [34]. These phenomena were demonstrated in different studies and resulted in prolonged tumor growth delay and tumor-free mice. In one of our studies, targeting tumor vasculature by silencing the expression of CD105 by GET resulted in both vascular disruption and prevention of the formation of new tumor vasculature. Furthermore, this resulted in a prolonged tumor growth delay (6 days) [32], which was also confirmed in this study, where in addition to silencing CD105, we also silenced CD146 simultaneously. CD146 is a marker involved in different signaling pathways; in the case of our study, it is important primarily as an alternative to CD105 in endothelial signaling. In our previous study, the efficacy of CD146 silencing was already evaluated in a TS/A tumor model, where a growth delay of 8.9 days was observed [33]. In our current study, we also obtained 12.5% tumor-free mice, which was not observed in our previous studies in this tumor model. However, the therapeutic effectiveness in the TS/A tumor model was still lower than that in the murine melanoma tumor model B16F10 using GET of plasmids silencing either CD105 or CD146 alone; specifically, silencing CD105 in melanoma resulted in pronounced tumor growth delay (8.6 days) and tumor-free mice (44%), from which the majority were also resistant to secondary challenge (75%) [30]. Silencing CD146 in the same melanoma tumor model resulted in a more pronounced tumor growth delay (13.2 days) and more tumor-free mice (35.7%), of which 60% remained tumor free after the secondary challenge [33]. These differences could be attributed to the different expression statuses of the targeted markers in specific tumor models, namely, while adenocarcinoma TS/A lacks the targeted markers CD105 and CD146, melanoma B16F10 expresses both of the markers; therefore, GET of the plasmid encoding either of these markers also has an impact on tumor cells itself, in addition to the antiangiogenic and vascular disrupting effects on endothelial cells in tumor vasculature.

Therapeutic effectiveness was reflected in the histological analyses of the tumors; the levels of vascularization and proliferation were decreased, while elevated levels were observed for hypoxia, apoptosis, necrosis and granzyme B-positive cells in the GET pDual group compared with the pertinent control group. The same phenomenon was observed in other aforementioned studies [30,32,33], again indicating the vascular-targeted effect of this treatment in the TS/A tumor model.

It is well known that improving the oxygenation status of the tumor can improve radiation therapy [34], which is also the case with silencing either of the markers CD105 or CD146 [30,33]. To improve the oxygenation status of tumors, we tested the gene silencing of two independent targets, CD105 and CD1045. While in the past we silenced each target separately, either using siRNA or improved treatment using GET of shRNA encoding plasmids, simultaneous targeting of the two mentioned genes was performed for the first time in the current study. For silencing, we used a novel double targeting plasmid that was prepared and tested in our previous in vitro study [25]. In the current study, we observed that a single 15 Gy dose had a moderate impact on proliferating tumor cells and tumor vasculature, and the tumor growth delay was not significant since the TS/A tumor model is known to be radioresistant [24,33,44]. However, IR combined with EP alone already resulted in a moderate but significant tumor growth delay (12.8 days) and 12.5% of tumor-free mice, which is also in accordance with our previous studies [29,30,33,45], confirming once again the efficacy of the EP on the radiosensitization status of tumors. By adding GET to the therapeutic plasmid, the effectiveness was significantly improved (37.3 days of tumor growth delay) in 50% of tumor-free mice, which is a significantly better outcome than that observed in previous studies by targeting either CD146 alone (28.7 days TGD and 26.7% CR) [33] or CD105 alone (15 days TGD and 40% CR) [29]. These observations were in line with histological analyses of tumors, where statistically significant reductions in vasculature (to 38.7%) and proliferating tumor cells (to 64.4%) were observed, while levels of hypoxia, apoptosis, necrosis and granzyme B-positive cells were statistically significantly elevated compared with the control group and monotherapies. These results indicate that GET of the novel dual tumor vasculature targeting plasmid is an improved treatment modality, which could have a great impact on future studies on targeting radioresistant tumor models.

One important observation of the current study is also the effectiveness of the GET of the control plasmid combined with irradiation (GET pEmpty + IR). Namely, compared with the control group and IR alone, it resulted in pronounced antitumor effectiveness, with a significant tumor growth delay (13.9 days) and 25% of tumor-free mice. The histological trends were more moderate than after GET of pDual and IR, but yet still followed the trend observed in the therapeutic group. These results were not completely unexpected, as similar results were already reported in other studies using GET of different control nontherapeutic plasmids, which resulted in up to 20% tumor-free mice in the TS/A tumor model [29,33]. In more immunogenic tumor models, such as melanoma, this effect was even more pronounced and resulted in up to 89% of tumor-free mice [30,33], which were mostly also resistant to secondary challenge. The observed therapeutic effectiveness after GET of the pEmpty plasmid correlated with increased expression of the DNA sensor Sting and pro-inflammatory cytokines. The rationale behind the effectiveness of the combination of GET and IR lies in the stimulation of the immune system through foreign DNA introduced into cells by gene therapy as well as DNA fragments released from damaged tumor cells after IR, which together act as PAMPs and DAMPs, activating DNA sensors [37,38,39,46]. Namely, DNA sensors are activated by the abnormal presence of DNA in the cytosol, and their activation leads to the upregulation of proinflammatory cytokines and different types of cell death [47]. In our previous study, we confirmed that cytokines and DNA sensors were upregulated in different tumor and nontumor cell types after GET of control plasmids and after IR of cells [36,37,48]. Stimulator of interferon genes (STING) is an important DNA sensor triggering type I interferons and proinflammatory cytokine production, and its importance in the tumor radiation response has been confirmed in numerous studies [11,12,13,14,44]. As expected, its levels were also significantly increased in our study, regardless of the plasmid that was used. Additionally, in the GET pEmpty + IR and GET pDual + IR groups, all tested cytokines (*Il1β, Ifn-β1* and *Tnf-α*) were upregulated 6 days after treatment, further confirming DNA sensor activation and adding to the therapeutic effectiveness of our treatment.

Taken together, we achieved significant tumor growth delay in the groups combining GET and IR; however, the outcome was better when the therapeutic plasmid was used since it also had a significant effect on the targeted tumor vasculature, therefore promoting radiosensitization of the tumors. Additionally, by combining GET and IR, which unselectively target vasculature and tumor cells, we were able to stimulate the immune system, even in the case of the control plasmid, and even further improve the therapeutic outcome. The results of this study are encouraging, since we obtained 50% tumor-free mice in an otherwise immunologically cold and radioresistant TS/A tumor model. Furthermore, we have even greater expectations for curing tumors that are immunologically hot; therefore, we believe that this therapy could be a promising novel treatment modality for a wide range of tumors.

## 4. Materials and Methods

### 4.1. Cell Lines

The mouse mammary adenocarcinoma cell line TS/A [49] (American Type Culture Collection, Manassas, VA, USA) was cultured in advanced DMEM (Dulbecco’s Modified Eagle Medium, Thermo Fisher Scientific, Waltham, MA, USA) supplemented with 5% fetal bovine serum (FBS), 10 mM/μL-glutamine (GlutaMAX) and 1% penicillin–streptomycin (all Thermo Fisher Scientific) in a 5% CO_2_ humidified incubator at 37 °C. Cells at 80% confluence were trypsinized using 0.25% trypsin/ethylenediaminetetraacetic acid in Hank’s buffer (Thermo Fisher Scientific), washed with Advanced DMEM with FBS, and collected by centrifugation (470 *g*, 5 min). The cell line was frequently tested for mycoplasma contamination by MycoAlertTM PLUS Mycoplasma Detection Kit (Lonza, Basel, Switzerland) and was confirmed to be mycoplasma free.

### 4.2. Plasmids

In the experiments, two plasmids constructed and evaluated in vitro by our research group were used [25]. Both were obtained by ORT technology to prepare antibiotic resistance gene-free plasmids. The therapeutic plasmid pDual (pU6-antiCD105-CD146-ORT) encodes two shRNAs against CD105 and CD146, while the plasmid pEmpty (pEmpty-ORT) has no homology to any gene in the mouse genome and was used as a control plasmid. The plasmids were isolated using an EndoFree Plasmid Mega Kit (Qiagen, Hilden, Germany) according to the manufacturer’s protocol. Plasmid purity and concentration were measured spectrophotometrically (Epoch microplate spectrophotometer, Take3 microvolume plate, BioTek). Additionally, the concentration and identity were confirmed by restriction analysis on an electrophoretic gel. The final concentrations were fine-tuned using a Qubit 4 Fluorometer (Thermo Fisher Scientific). The working concentration of 4 mg/mL was prepared in endotoxin-free water supplied within the kit.

### 4.3. Animals

Female BALB/cOlaHsd mice were purchased from Envigo RMS SrL (San Pietro al Natisone, Italy) and subjected to an adaptation period of 1 week. The mice were housed in individually ventilated cages and specific pathogen-free conditions at a temperature of 20–24 °C, relative humidity 55 ± 10%, and a 12 h light/dark cycle. Food and water were provided ad libitum. All procedures were performed in compliance with the guidelines for animal experiments of the EU directive (2010/63/EU) and permission from the Veterinary Administration of the Ministry of Agriculture and the Environment of the Republic of Slovenia (permission no. U34401-3/2022/11). The experiments were repeated twice, and the groups contained 6–8 animals per group.

### 4.4. In Vivo Gene Electrotransfer

The subcutaneous tumors were induced on the back of the mice by injecting 100 µL of 0.9% NaCl containing 2 × 10^6^ TS/A cells into BALB/cOlaHsd mice (one tumor per mouse). In vivo experiments were performed as described previously [29]. Briefly, when tumors reached 6 mm in the longest diameter, they were treated with an intratumoral injection (12.5 μL) of pDual (therapeutic plasmid) or pEmpty (control plasmid) or endotoxin-free water (mock control). Ten minutes after the injection, the electric pulses (EP) were delivered through two parallel stainless-steel electrodes; after the delivery of 4 pulses, the electrodes were turned 90° for the delivery of 4 additional pulses to expose the whole tumor. The parameters of EP were as follows: 8 square wave electric pulses with a voltage-to-distance ratio of 600 V/cm, a pulse duration of 5 ms, and a frequency of 1 Hz. The EP was generated by the electric pulse generator ELECTRO CELL B10 (Leroy Biotech, Saint-Orens-de-Gameville, France). The GET was performed on days 0, 2, and 4.

### 4.5. Tumor Irradiation

One day after the first GET, tumors were irradiated. The mice were placed in special lead holders with apertures for the local exposure of tumors. A single dose of 15 Gy at a dose rate of 1.92 Gy/min was delivered by a Glumay MP1-CP225 X-ray Generator (Gulmay Medical Ltd., Suwanee, GA, USA) operating at 200 kV and 9.2 mA with Cu (0.55 mm) and Al (1.8 mm) filtering.

### 4.6. The Tumor Growth Delay

The antitumor effect was determined by measuring three orthogonal diameters (a, b, c) of tumors using a Vernier caliper every second to third day. The tumor volume was calculated using the formula V = a × b × c × π/6. From the tumor volumes, arithmetic means for each group were calculated, and tumor growth curves were drawn with error bars representing the standard error of the mean. The tumor doubling time was defined as the time when tumors doubled in volume from the initial day of the experiment. The TGD was calculated as the difference in the tumor doubling times of the therapeutic and control groups. In all of the groups, tumor growth was followed until the tumors reached 350 mm^3^, which represented an event for the generation of Kaplan–Meier survival curves. Animals with tumors in the regression were examined weekly for tumor presence for 100 days after the treatment. The animals were considered cured or complete responders if they were tumor-free at day 100. The TGD of complete responders was set at 30 days for the generation of TGD graphs. The cured mice were challenged with a secondary subcutaneous injection of the tumor cells in the right flank as described above. Animals with no tumor growth 30 days after the injection of tumor cells were considered resistant to secondary challenge. Animal weight loss was monitored as a sign of systemic toxicity of the treatments. In addition, in the irradiated animals, acute skin reactions in the irradiated field were monitored as previously described [45].

### 4.7. Histology

From each experimental group, six tumors were collected on day 6 from the beginning of the experiment to evaluate their histological properties. The tumors were fixed in zinc fixative (BD Biosciences, San Diego, CA, USA), embedded in paraffin blocks, and cut into six consecutive 2-μm-thick sections for immunohistochemistry (IHC) analysis. The first section was stained with hematoxylin and eosin (H&E) to estimate the percentage necrotic tumor area, and the other five sections were stained immunohistochemically to determine the percentage of apoptosis, hypoxia, immune cells (granzyme B-positive cells), proliferation and the number of blood vessels.

Apoptosis was detected with antibodies against cleaved Caspase-3 (Ca-3, Cell Signaling Technology, Danvers, MA, USA) at a dilution of 1:500, and hypoxia was determined by hypoxia-inducible factor-1-α antibodies (ab2185, Abcam, Cambridge, MA, USA) at a dilution of 1:2000. For staining of immune cells (cytotoxic T lymphocytes and natural killer cells), antibodies against granzyme B (ab4059, Abcam) were diluted in a 1:1600 ratio. Proliferation was determined by antibodies against Ki-67 (clone SP6, Thermo Fisher Scientific) at a dilution of 1:1250. Blood vessels were visualized with antibodies against CD31 (ab28364, Abcam) at a dilution of 1:1000. Primary antibodies were detected with a peroxidase-conjugated streptavidin-biotin secondary antibody (Rabbit-specific HRP/DAB (ABC) detection IHC kit ab64261, Abcam) and counterstained with hematoxylin, as described previously [30]. The whole area of the tumor H&E-stained section was captured by a DP72 CCD camera connected to a BX-51 microscope (Olympus, Hamburg, Germany) under 10× magnification (numerical aperture 0.40). The necrotic area was evaluated by two independent researchers and presented as the percent of the necrotic area of the tumor section. From the remaining (five) immunohistochemically stained sections, at least five viable parts of each tumor sample were captured under 40× magnification (numerical aperture 0.85). The captured images were analyzed by two independent researchers and presented as the percentage of positive cells (apoptosis, hypoxia, immune cells, proliferation) or the number of blood vessels, as described previously [30].

### 4.8. Quantitative Reverse Transcription-Polymerase Chain Reaction (RT–qPCR)

From each experimental group, six tumors were collected on day 6 from the beginning of the experiment to determine the expression of the DNA sensor *Sting* and cytokines *Il1β*, *Ifn-β1* and *Tnfα*. The tumors were ground, and total RNA was isolated using TRIzol™ Reagent (Thermo Fischer Scientific) and purified with a peqGOLD Total RNA kit (PEQLAB, VWR™, Life Science, Leuven, Belgium) according to the manufacturer’s instructions. The concentrations and purity of RNA were quantified spectrophotometrically using a Cytation 1 Imaging Multi-Mode Reader (Agilent (BioTek instruments) Santa Clara, CA, USA). Total RNA (500 ng) was reverse transcribed into complementary DNA (cDNA) using a SuperScript VILO cDNA Synthesis Kit (Thermo Fisher Scientific). Tenfold diluted mixtures of transcribed cDNA were used as a template for RT–qPCR using PowerUp SYBR Green Master Mix (Thermo Fisher Scientific) and the primers (IDT, Newark, NJ, USA) specified in Appendix A. The reaction was performed on a QuantStudio™ 3 Real-Time PCR System (Thermo Fisher Scientific) under the cyclic conditions specified in Appendix A, and the results were analyzed with QuantStudio^®^ Design & Analysis Software v1.1 (Thermo Fisher Scientific). The expression was quantified using the ΔΔCt method [50] relative to the reference β-actin and glyceraldehyde 3-phosphate dehydrogenase mRNA and normalized to the control group.

### 4.9. Statistical Analysis

GraphPad Prism (GraphPad, San Diego, CA, USA) was used for statistical analyses. All data were tested for the normality of distribution with the Shapiro–Wilk test. Data are presented as the arithmetic mean (AM) ± the standard error of the mean (SEM). The differences between the experimental groups were statistically evaluated by one-way analysis of variance (one-way ANOVA) followed by Fisher’s LSD test for multiple comparisons. A P value of less than 0.05 was considered statistically significant. Survival was estimated by the Kaplan–Meier method, and survival curves were compared by the log-rank test. Tumor volumes of 350 mm^3^ were counted as events for the construction of the survival curves.

## Figures and Tables

**Figure 1 ijms-24-02755-f001:**
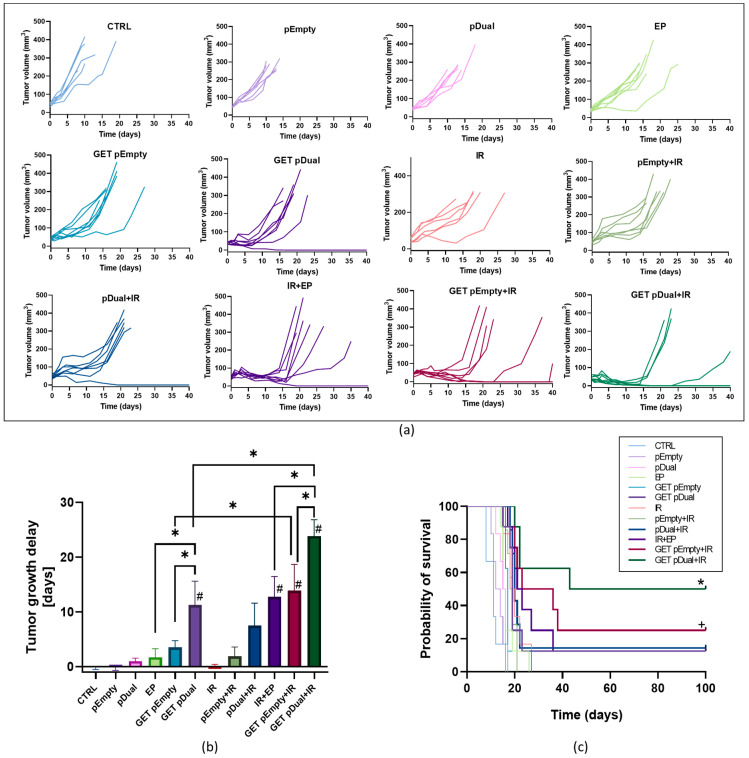
Therapeutic effectiveness in a murine mammary adenocarcinoma TS/A tumor model. (**a**) Individual tumor growth curves in all experimental groups. (**b**) Tumor growth delays in all experimental groups presented as the mean TGD with SEM. #, *p* < 0.05 vs. CTRL and IR; *, *p* < 0.05 vs. indicated group (only relevant comparisons are displayed: EP vs. GET pDual, EP vs. GET pEmpty, GET pEmpty vs. GET pDual, GET pEmpty vs. GET pEmpty + IR, GET pDual vs. GET pDual + IR, EP + IR vs. GET pEmpty, EP + IR vs. GET pDual, and GET pEmpty + IR vs. GET pDual + IR). (**c**) Kaplan–Meier survival curves with a tumor volume of 350 mm^3^ counted as an event. *, *p* < 0.05 vs. all groups except pDual, EP + IR and GET pEmpty + IR; +, *p* < 0.05 vs. all groups except EP + IR, GET pDual and GET pDual + IR. n = 6–8 mice.

**Figure 2 ijms-24-02755-f002:**
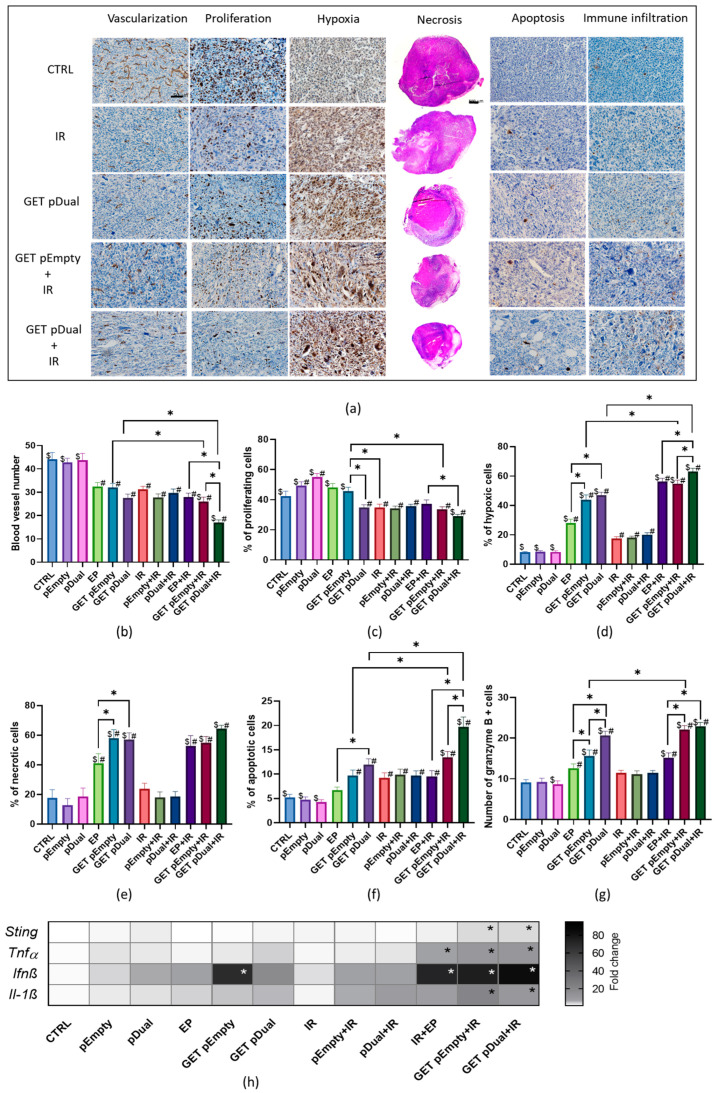
Histological analysis and expression of inflammatory cytokines and Sting. (**a**) Histological tumor sections on day 6 after the beginning of the therapy: vascularization, CD31 staining; proliferation, Ki-67 staining; hypoxia, hypoxia inducible factor-1-α staining; necrosis, HE staining; apoptosis, cleaved Caspase-3 staining; and immune infiltration, granzyme B staining. Scale bar in tumor section, 50 µm; scale bar in whole tumor area (necrosis), 500 µm. (**b**) Vascularization. (**c**) Proliferation. (**d**) Hypoxia. (**e**) Necrosis. (**f**) Apoptosis. (**g**) Immune infiltration. The data represent the arithmetic mean with the standard error of the mean per field of view; n = 6 tumors per group, of which at least 5 viable parts of the tumor were analyzed. #, *p* < 0.05 vs. CTRL; $, *p* < 0.05 vs. IR; *, *p* < 0.05 vs. indicated group (only relevant comparisons are displayed: EP vs. GET pEmpty, EP vs. GET pDual, GET pEmpty vs. GET pDual, EP + IR vs. GET pEmpty + IR, EP + IR vs. GET pDual + IR, and GET pEmpty + IR vs. GET pDual + IR). (**h**) Expression of cytokines Il1β, Ifn-β1 and Tnf-α and DNA sensor Sting on day 6 after the beginning of the therapy. *, *p* < 0.05 vs. CTRL. CTRL, control; GET, gene electrotransfer; IR irradiation.

**Table 1 ijms-24-02755-t001:** Tumor growth delay and complete responses in different experimental groups.

**Group**	**TGD (Days)**		**Complete Responders**		**Secondary Challenge**	
	n	AM ± SEM	n	%	n	%
CTRL	6	0.0 ± 0.5	0	0	/	/
pEmpty	6	−0.1 ± 0.6	0	0	/	/
pDual	6	1.0 ± 0.5	0	0	/	/
EP	8	1.7 ± 1.5	0	0	/	/
GET pEmpty	8	3.5 ± 1.2	0	0	/	/
GET pDual	8	11.3 ± 4.3	1	12.5	0	0
IR	6	0.0 ± 0.4	0	0	/	/
pEmpty + IR	7	1.9 ± 1.7	0	0	/	/
pDual + IR	7	7.5 ± 4.1	1	14.3	0	0
EP + IR	8	12.8 ± 3.7	1	12.5	0	0
GET pEmpty + IR	8	13.9 ± 4.8	2	25	0	0
GET pDual + IR	8	23.9 ± 3.0	4	50	1	25

AM, arithmetic mean; SEM, standard error of arithmetic mean; n, number of mice; TGD, tumor growth delay; CTRL, control; GET, gene electrotransfer; IR, irradiation.

## Data Availability

The data presented in this study are available on request from the corresponding authors.

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
