# Peer review of "Tumor Radiosensitization by Gene Electrotransfer-Mediated Double Targeting of Tumor Vasculature"

_ijms, 2023, doi:10.3390/ijms24032755_

Round 1

Reviewer 1 Report

In this study, the effects of gene electrotransfer-mediated delivery of a plasmid developed to silence CD105 (endoglin) and CD146 (melanoma cell adhesion molecule) was tested in combination with tumor irradiation of the non-immunogenic mammary adenocarcinoma in a syngeneic mouse model. The results show potential for the combination treatment with reduced growth rate and increased number of tumor-free mice compared to either treatment alone and accompanied by indications of increase in necrosis, hypoxia, apoptosis and infiltration of immune cells. Interestingly, ER with an empty plasmid in combination with IR also had an effect suggesting an immune response induced by cytoplasmic DNA corroborated by cytokine upregulation. Overall, this is a well conducted study.

Line 48: "mono- and combination-" should be "mono- or combination-"

Line 88: "one of important being the stimulator of interferon genes (STING)".  Should be "importance"

p.3 line 107: EP must be defined at first use

Line 116: "where there were also some complete responders that were tumor-free" Where not all complete responders tumor-free?

p.10 l 382 "Cured mice were challenged with a secondary subcutaneous injection of the tumor cells as described above in the right flank" should be "Cured mice were challenged with a secondary subcutaneous injection of the tumor cells in the right flank as described above ".

Figure 1a: The number of curves do not correlate with the number of mice in table 1 for each group and where are the complete responders? If the complete responders are not included in the figure, this should be stated in the legend. But still, GET pEmpty+IR only has 5 curves out of 8 mice and 2 complete responders and EP+IR 5 curves out of 8 mice and 1 complete responder?

Figure legend 1b: "vs. CTRL and IR" should be "vs. CTRL or IR"

There are no references to figure 2 in the text

Author Response

Dear reviewer,

Thank you for the fast review and useful comments and suggestions. Below are the point by point responses to the comments, which we also addressed in the revised version of the manuscript in the attachment.

Kind regards,

Urška Kamenšek

  1. Line 48: "mono- and combination-" should be "mono- or combination-"

The sentence was corrected as suggested.

  1. Line 88: "one of important being the stimulator of interferon genes (STING)".  Should be "importance"

The sentence was corrected as suggested.

  1. 3 line 107: EP must be defined at first use

We apologise for this mistake, which we corrected in the revised version.

  1. Line 116: "where there were also some complete responders that were tumor-free" Where not all complete responders tumor-free?

Thank you for noticing this lapsus. Of course, all complete responders were tumor- and also metastasis-free (please see also the response to 6th comment). The sentence was corrected according.

  1. 10 l 382 "Cured mice were challenged with a secondary subcutaneous injection of the tumor cells as described above in the right flank" should be "Cured mice were challenged with a secondary subcutaneous injection of the tumor cells in the right flank as described above ".

The sentence was corrected as suggested.

  1. Figure 1a: The number of curves do not correlate with the number of mice in table 1 for each group and where are the complete responders? If the complete responders are not included in the figure, this should be stated in the legend. But still, GET pEmpty+IR only has 5 curves out of 8 mice and 2 complete responders and EP+IR 5 curves out of 8 mice and 1 complete responder?

We apologize for this. The mistake in group GET pEmpty+IR accrued because one of the mice was tumor free, however she developed lung metastases at day 39. We don’t know what happened in the group EP+IR, where 2 mice were accidentally lost in the GraphPad file at some point. Thank you for noticing. To be more transparent, we decided to draw curves for all mice, including the complete responder. We also prolonged the x axis so that the mouse that developed the metastases could be seen.

  1. Figure legend 1b: "vs. CTRL and IR" should be "vs. CTRL or IR"

The figure legend was corrected as suggested.

  1. There are no references to figure 2 in the text

Thank you very much for noticing this mistake. As the second reviewer also noticed this and suggested that we combine the figure 2 with figure 3, we did so. According we also corrected all the references to newly marked figure.

Reviewer 2 Report

In the current manuscript, entitled “Tumor radiosensitization by gene electrotransfer-mediated double targeting of tumor vasculature”, Savarin et al., showed that therapeutic potential of knock down CD105 and CD1046 can be synergized with IR in mammary adenocarcinoma tumor model TS/A. Further, they performed histological analysis of tumors to validate the therapeutic effect. Finally, they discovered elevation of DNA sensor STING and pro-inflammatory cytokines Il1β, Ifn-β1 and Tnf-α by qRT-PCR.

In general, the conclusion can be supported by the data. The authors introduced the story in a logic way. The manuscript is well written. I only have some minor points.

1 What is EP?Should be defined in line110.

2 Please define the tumor cell type they used in Fig1.

3 Fig3a-f should be combined with Fig2. Because quantification follow the image would be much easier to understand.

4 Please define how many tumors they induced in each mice.

Author Response

Dear reviewer,

Thank you for the fast review and useful comments and suggestions. Below are the point by point responses to the comments, which we also addressed in the revised version of the manuscript in the attachment.

Kind regards,

Urška Kamenšek

  1. What is EP?Should be defined in line110.

We apologise for this mistake, which we corrected in the revised version.

  1. Please define the tumor cell type they used in Fig1.

The tumor model was defined in the figure legend.

  1. Fig3a-f should be combined with Fig2. Because quantification follow the image would be much easier to understand.

Thank you for your useful suggestion. The two figures were combined and all the references to newly marked figure were corrected accordingly.

  1. Please define how many tumors they induced in each mice.

The number of tumors per mouse, i.e. 1, was defined in the M&M section.
